# Host environment shapes filarial parasite fitness and *Wolbachia* endosymbionts dynamics

Frédéric Fercoq[1], Clément Cormerais[1], Estelle Remion[1], Joséphine Gal[1], Julien Plisson[2], Arame Fall[2], Joy Alonso[1], Nathaly Lhermitte-Vallarino[1], Marc P. Hübner[3,4], Linda Kohl[1], Frédéric Landmann[2☉], Coralie Martin[1☉*]

1 Unité Molécules de Communication et Adaptation des Micro-organismes (MCAM, UMR 7245), Muséum National d'Histoire Naturelle, CNRS, Paris, France, 2 Centre de Recherche en Biologie Cellulaire de Montpellier (CRBM), Université de Montpellier, CNRS, Montpellier, France, 3 Institute for Medical Microbiology, Immunology & Parasitology (IMMIP), University Hospital of Bonn, Bonn, Germany, 4 German Center for Infection Research (DZIF), Partner Site Bonn-Cologne, Bonn, Germany

☉ These authors contributed equally to this work.
* coralie.martin@mnhn.fr

## Abstract

Filarial nematodes, responsible for diseases like lymphatic filariasis and onchocerciasis, depend on symbiotic *Wolbachia* bacteria for reproduction and development. Using the *Litomosoides sigmodontis* rodent model, we investigated how host type-2 immunity influences Wolbachia dynamics and parasite development. Wild-type and type-2 immune-deficient (*Il4rα⁻/⁻Il5⁻/⁻*) BALB/c mice were infected with *L. sigmodontis*, and the distribution and abundance of *Wolbachia* were analyzed at different developmental stages using quantitative PCR and fluorescence *in situ* hybridization. Our results show that type-2 immune environments selectively reduce germline *Wolbachia* in female filariae from wild-type mice, a change associated with disrupted oogenesis, embryogenesis, and microfilarial production, while somatic *Wolbachia* remain unaffected. Antibiotic treatments achieving systemic *Wolbachia* clearance result in similar reproductive impairments. Notably, *Wolbachia*-free microfilariae are observed shortly after *Wolbachia* depletion, suggesting that early-stage embryogenesis can proceed temporarily before progressive germline dysfunction ensues. *Wolbachia*-free microfilariae develop into infective larvae in the vector, but stall beyond the L4 stage in vertebrate hosts, showing arrested growth and reproductive organ maturation defects in both male and female larvae. These findings highlight the variable dependency on *Wolbachia* across life stages and provide insights into host-parasite-endosymbiont interactions shaped by environmental pressures.

## Author summary

Filarial parasites are responsible for debilitating diseases like lymphatic filariasis and onchocerciasis, which affect millions of people worldwide, causing severe

**Data availability statement:** All data generated or analyzed during this study are included in this published article and its supporting information files. The underlying numerical values are provided in S1 Table.

**Funding:** This research was supported by core funding from the Muséum National d'Histoire Naturelle, the French Agence Nationale de la Recherche (ANR) grant Project WOLF (ANR-21-CE13-0029) awarded to CM and FL, and the Actions Thématiques du Muséum (ATM) awarded to FF. The funders had no role in study design, data collection and analysis, decision to publish, or preparation of the manuscript.

**Competing interests:** The authors have declared that no competing interests exist

disability and social stigma. These parasites rely on symbiotic bacteria of the genus *Wolbachia* for their growth and reproduction. Because of this relationship, *Wolbachia* have become a key target for treatments aiming at reducing parasite survival and transmission. In our study, we discovered that the host's immune system can influence this parasite-bacteria interaction. We found that type-2 immune responses are associated with a reduction of *Wolbachia* in the reproductive organs of the parasites, coinciding with fewer offspring and disrupted development. Interestingly, while some parasites could still produce offspring without *Wolbachia*, these offspring failed to fully mature, which limits the parasites' ability to spread. These findings reveal how the immune system can interfere with the life cycle of filarial parasites by affecting their bacterial partners. This insight could help improve treatments that target *Wolbachia*, offering new strategies to control filarial diseases and reduce their impact on public health.

## Introduction

Filarial nematodes from the Onchocercidae family are parasites that infect terrestrial vertebrates [1]. Filariae are transmitted by blood-feeding arthropods and are responsible for debilitating diseases, collectively known as filariasis, affecting over 100 million people today [2]. In species infecting humans, filariae typically live for 10–15 years, remaining fertile for 5–8 years and producing millions of offspring [2]. These diseases include Lymphatic Filariasis (LF) and Onchocerciasis, for which no safe short-term treatments that eliminate the adult worms exist [3]. Filarial infections also threaten animals like dogs (heartworm disease) and cattle [4–6], driving ongoing veterinary research to combat these parasites [7]. Filarial nematodes share a life cycle with four larval stages (L1–L4) and an adult stage comprising male and female worms [8]. Adult filariae exhibit a high degree of tissue specificity in their vertebrate hosts, residing in different parts of the body such as subcutaneous tissue, lymphatics, blood, or pleural and peritoneal cavities, depending on the filarial species.

Among approximately a hundred species of filarial nematodes analyzed, more than 50% maintain a mutualistic relationship with the alpha proteobacteria *Wolbachia* [9–11]. *Wolbachia* are considered the most successful endosymbionts on earth, as they infect most terrestrial arthropods in addition to Onchocercidae [11,12]. *Wolbachia* endosymbionts are involved in developmental processes that regulate female germline proliferation, and they are essential for maintaining spatial organization of early oogenesis in the distal ovary [13]. *Wolbachia* also play a key role in modulating the host's immune responses, particularly by interfering with type 2-mediated mechanisms essential for parasite clearance [14]. For instance *Wolbachia* endosymbionts play a critical role in modulating the host immune response in *Onchocerca* nodules [15–17]. These bacteria induce the recruitment of neutrophils preventing eosinophil responses, which are essential for the clearance of the parasite [15]. After *Wolbachia* depletion via antibiotics like tetracyclines, neutrophils decrease, and eosinophils infiltrate the nodules, release their granules on the worm cuticle and induce parasite

death [15,16]. Antibiotic-induced depletion of *Wolbachia* significantly reduces the lifespan of filarial nematodes and rapidly sterilizes them [18–20]. However, the mechanisms governing the interactions between the endosymbiont and the parasite are still unknown.

In human filarial infections, the balance of T helper cell responses strongly influences disease outcomes. Asymptomatic individuals typically mount regulated Th2 responses, associated with parasite control and tissue repair, while patients with chronic pathology, such as lymphedema, display increased Th1 and Th17 cytokine production and reduced regulatory T cell activity [21–24]. While these responses clearly influence host resistance, their role in regulating the filaria–*Wolbachia* symbiosis remains unexplored.

Immune responses to filariae have been extensively analyzed using the *Litomosoides sigmodontis* murine model of filariasis [25–28]. In *Rag2IL-2Rγ$^{-/-}$* C57BL/6 mice, the absence of adaptive immune responses leads to a significant increase in *Wolbachia* bacterial load within adult female worms, which correlates with an enhanced parasite survival and fertility [29]. In the relatively susceptible BALB/c strain, infection of mice lacking of eosinophils (IL-5 knockout or Δ*dblGata1* mice) increases parasite survival and filarial embryogenesis compared to wild-type mice [30–34]. Additional depletion of IL-4Rα signaling (*Il4ra$^{-/-}$/Il5$^{-/-}$* mice) increases the susceptibility, leading to enhanced production of microfilariae and a significantly increased worm survival [33–36]; however the specific consequences for *Wolbachia* remain unclear [35]. This increased susceptibility in type 2-deficient *Il4ra$^{-/-}$/Il5$^{-/-}$* mice correlates with impaired activation of alternative macrophages, which are unable to fully mature into functional resident macrophages [36–38]. These observations suggest that filarial developmental defects may be either directly caused by *Wolbachia* being targeted by type 2 immune responses, or an indirect consequence of endocrine or metabolic environments modified by these genetic alterations.

Understanding how *Wolbachia* interact with their nematode host requires a good knowledge of the anatomical context beyond their tissue-specific localization within filariae [12,39]. Filarial nematodes exhibit bilateral symmetry, with their bodies divided into two equal halves along a central plane. Beneath their protective cuticle lies the hypodermal layer, which plays a key role in secreting and maintaining the cuticle [40] as well as absorbing nutrients and metabolites [39]. In adult worms, the hypodermis forms a continuous syncytial layer, which is thickened dorsally, ventrally, and laterally to form four prominent hypodermal cords [39]. Among these, the two lateral hypodermal cords host *Wolbachia*, inherited from embryonic hypodermal precursor cells [41]. After the third molt (L3 to L4), these somatic *Wolbachia* migrate in a cell-to-cell manner to colonize the female germline, where they support the growth and maturation of oocytes [13,41]. Within the pair of tube-shaped ovaries, *Wolbachia* utilize the central cytoplasmic rachis of the syncytial germline to distribute themselves throughout developing oocytes prior to cellularization, ensuring vertical transmission. First, daughters of distal germline stem cells divide mitotically along the proliferative zone (PZ), where *Wolbachia* enhance proliferation and maturation presumably with the help of distal sheath cells, that have been shown to participate to mitotic proliferation regulation in *C. elegans* [13,42]. Subsequently, germ cell nuclei undergo a premeiotic S phase and differentiate in the meiotic zone (MZ), eventually resulting in competent oocytes after cellularization [13,43]. These oocytes are then released in the pair of uteri where they become fertilized and undergo embryogenesis to form microfilariae, the motile larvae being released in the host [20,39,41,44]. The symbiosis between *Wolbachia* and filarial nematodes is essential for their reproductive success. Disruption of this symbiosis, for example by antibiotic treatment, leads to major reproductive failures, starting with germline defects worsening over time and eventually leading to aborted embryogenesis [13,20,41,44,45]. Enhanced worm reproduction observed in type 2-deficient *Il4ra$^{-/-}$/Il5$^{-/-}$* mice suggests that these responses may disrupt *Wolbachia* populations, impairing the parasites' reproductive capabilities. The primary aim of this study is to explore how a type 2 immune environment affect both *Wolbachia* density and localization within *L. sigmodontis* and the nematode's development and reproduction. By comparing worms raised in type 2-deficient (*Il4ra$^{-/-}$/Il5$^{-/-}$*) *versus* immune-competent (wild type) mice, we aim to elucidate how host immune mechanisms influence the bacterial endosymbionts, parasite fitness and reproductive success.

Our findings show that type 2 immune responses do not affect somatic *Wolbachia* populations, but create an unfavorable host environment for germline maintenance, indirectly impacting ovarian *Wolbachia* densities. Reciprocally, this depletion impairs oogenesis, embryogenesis, and microfilarial production. Strikingly, *Wolbachia*-free microfilariae produced shortly after antibiotic treatment, *i.e.*, before female sterility, do develop into infective larvae within the vector but fail to mature beyond the L4 stage in vertebrate hosts. These results highlight the stage-specific roles of *Wolbachia* in parasite reproduction and development, illustrating the intricate interplay between host environment, parasitic nematodes, and their symbiotic bacteria.

## Results

### Bacterial density in filarial parasites depends on the host's type 2 immune environment

To assess the impact of type 2 immune responses on *Wolbachia* in filariae, we conducted a kinetic study of *Wolbachia* titers in *Litomosoides sigmodontis* female worms from immune-competent wild-type (WT) and type 2-deficient *Il4rα⁻/⁻/Il5⁻/⁻* (KO) BALB/c mice. Parasites were collected before the final molt (late L4 larvae at 24- and 28-days post-infection [dpi]) and after the final molt (from immature young adults at 32 dpi to mature adults at 70 dpi). To evaluate *Wolbachia* titers, we measured the bacterial single-copy gene *ftsZ* by quantitative PCR (qPCR), using the ratio of *Wolbachia ftsZ* to *L. sigmodontis actin* as a measure of relative bacterial abundance in filarial worms [46].

In WT mice, the *ftsZ/actin* ratio increased slightly during worm development from L4 larvae to young adults (32 dpi), then decreased slowly (Fig 1A). Conversely, in filariae developing in *Il4rα⁻/⁻/Il5⁻/⁻* mice, our results revealed a significant peak in the *ftsZ/actin* ratio at 32 dpi, indicating a substantial increase in *Wolbachia* population following the molt to the adult stage. This peak was followed by a notable decline in the ratio between 50–70 dpi (Fig 1A). Importantly, this reduction in the *ftsZ/actin* ratio in filariae from *Il4rα⁻/⁻/Il5⁻/⁻* mice was not due to bacterial loss, as absolute quantification of *Wolbachia ftsZ* (Fig 1B) showed stable bacterial loads after 32 dpi onward. Notably, while the *ftsZ/actin* ratios were similar in

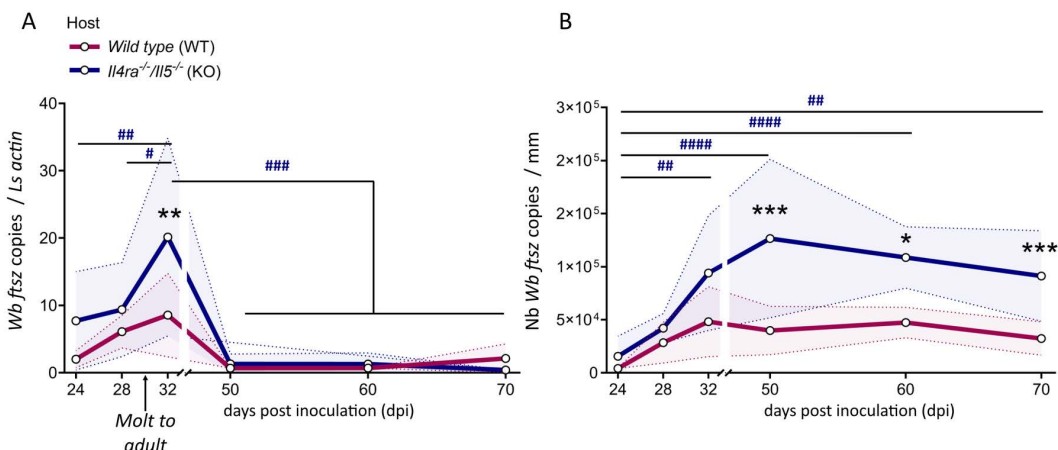

**Fig 1. A type 2 immune environment is associated with low *Wolbachia* titers in adult female filariae.** Type 2-competent (wild-type) and type 2-deficient (*Il4rα⁻/⁻/Il5*) were inoculated with 40 infective larvae (L3) of the filaria *Litomosoides sigmodontis*. Parasites were harvested at various time points before and after the fourth molt (indicated by the arrow at approximately 30 days post-infection, [dpi]), and *Wolbachia* density was evaluated in female filariae by qPCR of the bacterial gene *ftsZ*. **(A)** Relative quantification of *Wolbachia* density (ratio between *Wolbachia's ftsZ* and filarial *actin* gene copies) in female filariae (see S1 Fig for *actin* quantification). **(B)** Absolute quantification of *Wolbachia's ftsZ* counts in female filariae; results were normalized by average worm size in each group (see S1 Fig for worm sizes). Results are expressed as the ± SD of n = 4–6 filariae per group (24–60 dpi), n = 10–15 filariae per group (70 dpi). Two-way ANOVAs followed by Bonferroni's multiple comparisons tests were performed; *$p < 0.05$, **$p < 0.01$, ***$p < 0.001$ indicate significant difference between filariae from wild-type and *Il4rα⁻/⁻/Il5⁻/⁻* hosts. #$p < 0.05$, ##$p < 0.01$, ###$p < 0.001$, indicate significant differences between timepoints within the *Il4rα⁻/⁻/Il5⁻/⁻* group.

filariae from *Il4rα⁻/⁻/Il5⁻/⁻* and WT mice at 50–70 dpi, the absolute abundance of *Wolbachia* was significantly higher in *Il4rα⁻/⁻/Il5⁻/⁻* mice. This remained true after normalizing *ftsZ* copies to worm length to account for size differences, as adult worms from *Il4rα⁻/⁻/Il5⁻/⁻* mice are longer (9.91 ± 1.86 cm *vs* 8.23 ± 2.032 cm at 70 dpi; mean ± SEM, see S1A Fig and references [33–35]).

These findings suggest that a substantial increase in *Wolbachia* population occurs after the final molt in *Il4rα⁻/⁻/Il5⁻/⁻* mice, concurrent with parasite growth (S1A Fig) and an increase in *actin* copies (S1B Fig). In WT mice, type 2 immune responses appear to inhibit bacterial growth, reducing *Wolbachia* load in adult filariae and supporting the hypothesis that type 2 immunity limits bacterial proliferation.

**Both selective germline *Wolbachia* depletion by type 2 immune environment and systemic clearance by antibiotic regimen are associated with impaired filarial reproduction**

To determine whether *Wolbachia* depletion alone can limit filarial growth and fertility independently of type 2 immune mechanisms, we treated *Il4rα⁻/⁻/Il5⁻/⁻* (KO) mice with a doxycycline-rifampicin (DR) regimen. This approach allowed us to test whether the fertility reduction observed in WT mice could be replicated by directly targeting *Wolbachia*, even in the absence of type 2 immunity. The treatment consisted of a 14-day course starting at 40 days post-infection (dpi), therefore targeting mature adult worms. It was optimized from previous studies on mice [46,47] to induce a > 99% reduction in *Wolbachia* (Fig 2A), while maintaining parasite survival at 70 dpi (S2A Fig). Treated worms, however, were smaller, resembling worms from WT mice (S2B Fig).

Blood microfilaremia levels were monitored over time (Fig 2B). As previously described [25,26,33,34,36], wild type (WT) mice displayed moderate microfilaremia peaking at 70 dpi before gradually decreasing. In contrast, KO mice exhibited sustained high microfilaremia up to 150 dpi. DR-treated KO mice showed a similar early profile to untreated KO mice, but after 70 dpi, microfilaremia dropped to almost 0 around 120 dpi, similar to WT hosts. Hence, the embryos already developing during the treatment were able to successfully complete their embryogenesis in the absence of *Wolbachia* while embryos derived from depleted oocytes mostly lead to unviable embryos. This suggests that *Wolbachia* are crucial for early reproductive process but not for later stages of embryogenesis such as morphogenesis.

To further explore the link between *Wolbachia* and reproductive fitness, we next examined proximal uterine contents of whole-mount filariae at 55 and 70 dpi using fluorescence *in situ* hybridization (FISH) to visualize *Wolbachia* (Fig 2C and 2D; see S3A and S3B Fig for anatomical context). In worms from WT mice, the uteri contained a mixture of aborted embryos and microfilariae, including both *Wolbachia*-positive (*Wb*(+)) and *Wolbachia*-negative (*Wb*(-)) microfilariae. In contrast, worms harvested from KO mice had uteri densely packed with *Wb*(+) microfilariae, in accordance with the sustained microfilaremia observed in the blood. Worms from DR-treated mice initially mirrored the ones from untreated KO mice at 55 dpi, with viable but *Wolbachia*-depleted (*Wb*(-)) microfilariae. By 70 dpi, however, uterine contents in *Wolbachia*-depleted filariae decreased dramatically, resembling the phenotype observed in WT hosts, with predominantly aborted embryos and reduced microfilarial output. This coincided with the significant drop in blood microfilaremia levels, indicating a progressive worsening of the reproductive defects over time (Fig 2F and 2G).

Next, we assessed *Wolbachia* levels in the lateral hypodermal chords, which are the somatic tissues where *Wolbachia* also reside. Worms from both WT and KO mice displayed similar *Wolbachia* loads in the lateral chords (Fig 2E and 2H), suggesting that the environment associated with type 2 responses reduces bacterial density in reproductive tissues only. This highlights the selective impact of a type 2 genetic background, which affects *Wolbachia* in reproductive tissues, unlike the uniform bacterial depletion achieved by antibiotics.

We then investigated how the host's type 2 immune environment *versus* antibiotic treatment influence *Wolbachia* colonization and oogenesis in the ovaries of *L. sigmodontis* female filariae. The ovary of *L. sigmodontis* begins as a syncytium organized around a central cytoplasmic core (the rachis) surrounded by a single layer of somatic sheath cells providing structural support. *Wolbachia* are distributed along the rachis and within distal somatic sheath cells in the proliferative

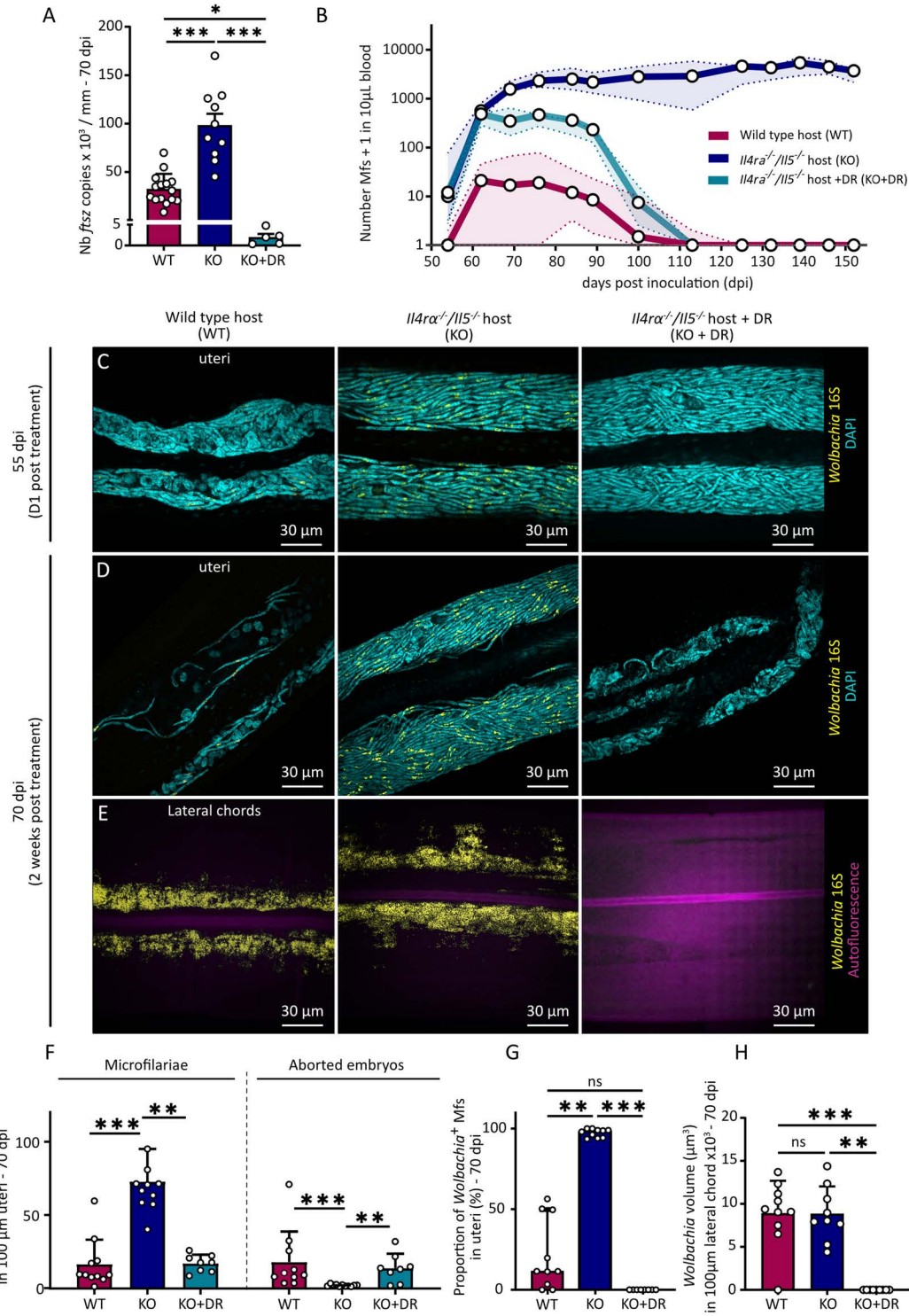

**Fig 2. Selective germline *Wolbachia* depletion by type 2 immunity and systemic antibiotic clearance are both associated with impaired filarial reproduction.** Wild type (WT) and *Il-4rα⁻/⁻/Il-5⁻/⁻* (KO) BALB/c mice were infected with 40 *L. sigmodontis* L3 larvae. From 40 dpi, KO mice were treated with a combination of rifampicin (10 mg/kg/day for 5 days) and doxycycline (100 mg/kg/day for 14 days) to deplete *Wolbachia* in filariae (KO+DR). **(A)** Absolute quantification of *Wolbachia's ftsZ* in female filariae; results were normalized by average worm size in each group. Results are expressed as the mean ± SEM of n = 15 filariae from WT mice, 10 from KO mice and 5 from KO+DR mice. **(B)** Kinetics of microfilaremia in blood from WT, KO and

KO + DR mice. Data points represent mean microfilarial counts per 10 μL of blood at different time points post-infection (dpi). Results are expressed as mean ± SEM for n = 3-23 mice per group and per timepoint. **(C-E)** Parasites were harvested at 55 and 70 dpi and whole mount female filariae were stained for DNA (DAPI, cyan) and *Wolbachia* (16S ribosomal subunit, yellow) for analysis by confocal fluorescence imaging. **(C-D)** Representative confocal images of the proximal uteri of female filariae from WT, KO and KO + DR mice at 55 dpi (C) and 70 dpi (D). **(E)** Representative confocal z-stack images of the lateral hypodermal chords of female filariae from WT, KO and KO + DR mice at 70 dpi. **(F)** Quantification of the number of microfilariae (left) and aborted embryos (right) in the uteri images, normalized to 100 μm. **(G)** Proportion (%) of *Wolbachia*-positive microfilariae in the uteri. **(F-G)** Results are expressed as mean ± SD of n = 11 filariae from WT mice, 11 filariae from KO mice and 8 filariae from KO + DR mice. Kruskal-Wallis tests were performed followed by Dunn's multiple comparison test. ns: no statistical difference, ** $p < 0.01$, *** $p < 0.001$. **(H)** Quantification of *Wolbachia* density in images of the proximal lateral chords (expressed in $\mu m^3/100$ μm) in 3D images. Results are expressed as mean ± SEM of n = 10 filariae from WT mice and n = 9 filariae from KO mice and 6 from KO + DR mice. A one-way ANOVA was performed followed by Tukey's multiple comparison test, *** $p < 0.001$.

zone (PZ), a distal zone defined as extending from the ampulla (the distal tip of the ovary) to the last observed mitotic nucleus ([13] and S3B and S3C Fig).

We imaged the PZ to investigate simultaneously germline proliferation, *Wolbachia* distribution and apoptosis (Fig 3A). Phospho-Histone 3 (PH3) staining revealed mitotic nuclei, FISH 16S probes labeled *Wolbachia*, and DAPI staining identified pyknotic nuclei (DAPI bright foci, PH3-negative) indicative of cellular apoptosis. Quantification of PH3-positive mitotic nuclei revealed enhanced germline proliferation in worms harvested from KO mice compared to WT mice (Fig 3B). Antibiotic-treated worms from KO + DR mice showed a marked collapse in mitotic proliferation to levels similar to worms from WT mice (Fig 3B). *Wolbachia* density was substantially lower in worms from WT mice than from KO mice, and endosymbionts were almost eradicated from filariae from KO + DR mice (Fig 3C). Concomitantly, apoptosis was significantly higher in worms from WT and KO + DR mice, contrasting sharply with the minimal apoptosis observed in worms from KO mice (Fig 3D).

Spatial analysis of the PZ images (quantifications at 100 μm intervals) revealed distinct patterns of mitotic activity and *Wolbachia* density across the three environments (Fig 3E and 3F). In worms developing in WT mice, the PZ was short (2.87 ± 0.53 mm, mean ± SD), with scattered PH3 + nuclei and a modest peak towards the end of the PZ. *Wolbachia* were present in both the rachis and the surrounding sheath cells, but their density gradually declined toward the end of the PZ. In filariae from KO mice, the PZ was significantly longer (3.68 ± 0.19 mm, mean ± SD), with mitotic nuclei concentrated at the end of the PZ, reflecting a robust germline amplification in transit. *Wolbachia* were also more abundant in worms raised in KO mice, densely colonizing both the rachis and the sheath cells, without the distal-to-proximal decline observed in a WT environment. In the KO + DR condition, mitotic activity was sparse and scattered (Fig 3A and 3E), with *Wolbachia* occasionally appearing as small residual patches in the rachis or sheath cells (Fig 3A and 3F).

High resolution imaging of the distal tip of the ovary (the ampulla) (Fig 3G, see S3 Fig for a diagram of ovary anatomy) revealed striking differences in *Wolbachia* titer and germline integrity, consistent with patterns observed throughout the PZ. In a WT mouse environment, *Wolbachia* density was reduced compared to KO mouse-derived worms in both the rachis and the surrounding somatic sheath cells and apoptotic germ cells were frequently observed. By contrast, in worms from KO mice, *Wolbachia* signal was intense and widespread, with strong bacterial titer throughout the rachis and extending into the sheath cells which appeared enlarged. Apoptotic nuclei were rarely detected in the ampulla. In filariae raised in KO + DR mice, *Wolbachia* were either entirely absent or present as isolated residual patches in the rachis or sheath cells. This indicates that *Wolbachia* titers and germline integrity in the ampulla can serve as a proxy for broader trends observed throughout the PZ.

Together, these findings suggest that both a type 2 immune environment and antibiotic treatment unbalance the intricate interplay between germline function and *Wolbachia* in *L. sigmodontis* female worms. Disrupting this balance leads to impaired oogenesis, likely producing faulty oocytes unable to proceed through embryogenesis. However, embryos originating from competent oocytes—formed before treatment or from partially functional germlines in a type 2 immune environment—can complete embryogenesis and give rise to viable microfilariae, even in the absence of *Wolbachia*.

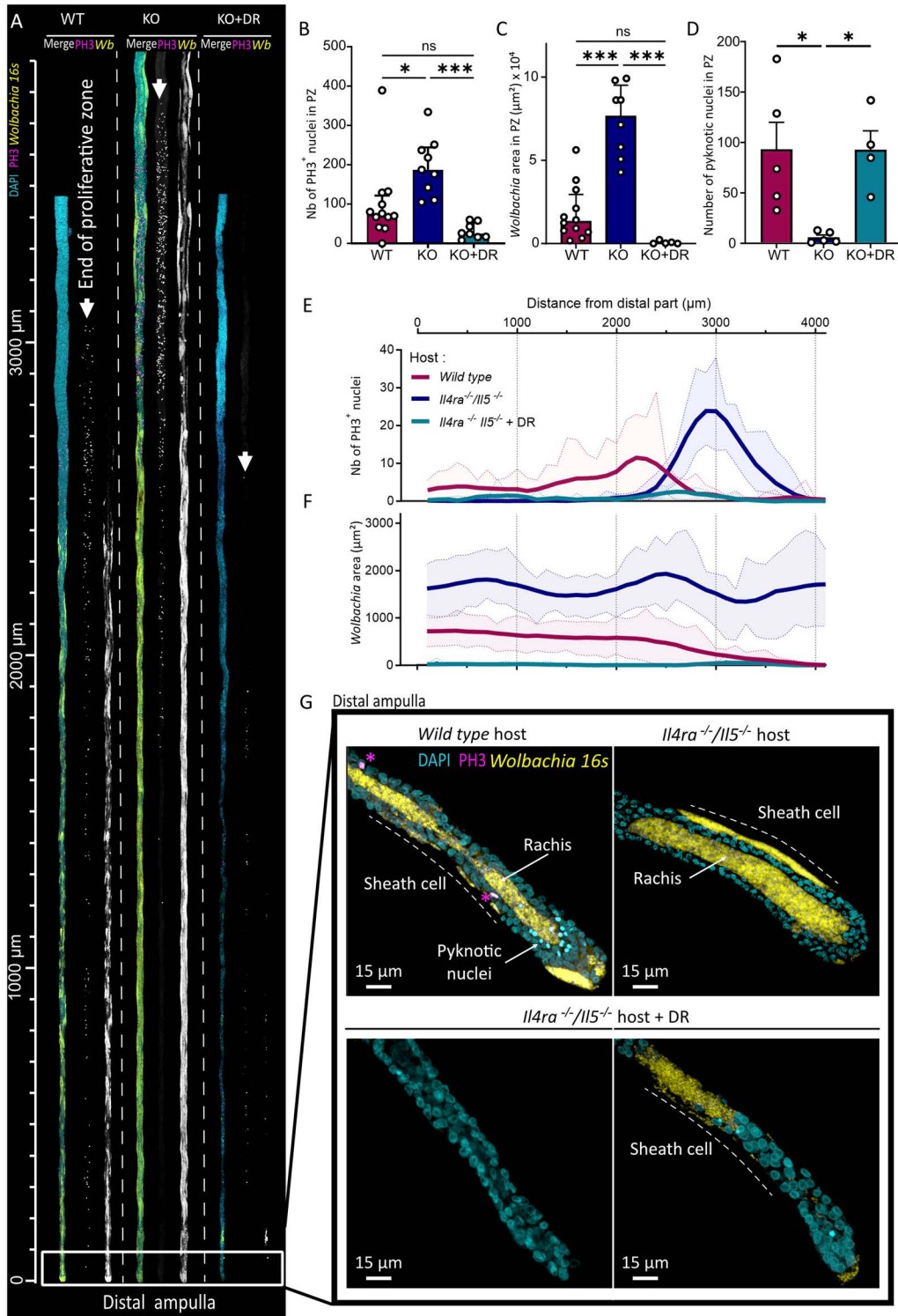

**Fig 3. Host genetic background and antibiotic-induced *Wolbachia* depletion disrupt filarial ovarian dynamics and oogenesis.** Wild-type (WT) and *Il4ra⁻/⁻/Il5⁻/⁻* (KO) BALB/c mice were infected with 40 *L. sigmodontis* L3 larvae. From 40 days post-infection (dpi), KO mice were treated with a combination of rifampicin (10 mg/kg/day for 5 days) and doxycycline (100 mg/kg/day for 14 days) to deplete *Wolbachia* (*KO+DR*). At 70 dpi, female parasites were harvested to analyze *Wolbachia* dynamics, oogenesis, and apoptosis in the filarial germline. Parasite's ovaries were dissected and stained for DNA

(DAPI, cyan), mitotic nuclei (Phospho Histone 3 – PH3, magenta), and *Wolbachia* (16S ribosomal subunit, yellow), and analyzed by confocal fluorescence imaging. **(A)** Linearized confocal images of the PZ in the distal part of filarial ovaries, showing DNA (DAPI, cyan), mitotic nuclei (PH3, magenta), and *Wolbachia* (yellow) signals. The PZ extends from the ampulla to the last PH3-positive nucleus (indicated by arrowheads). **(B)** Quantification of the total number of PH3 + nuclei in the PZ. Results are expressed as mean ± SD of n = 8-12 ovaries per group. **(C)** Quantification of *Wolbachia*+ coverage in the PZ. Results are expressed as mean ± SD of n = 5-12 ovaries per group. **(D)** Quantification of pyknotic nuclei (indicative of germline apoptosis) in the PZ. Results are expressed as mean ± SD of n = 5 ovaries per group. **(B-D)** Kruskal-Wallis tests were performed followed by Dunn's multiple comparison test. ns = not significant, *p < 0.05, ***p < 0.001. **(E-F)** Spatial analysis of PH3 + mitotic nuclei (E) and *Wolbachia*+ area (F) along the ovarian PZ. Results were segmented into 100-µm intervals, with mean data smoothed for 4 neighboring segments and standard deviations (SD) shown for raw data. n = 5-12 ovaries per group. **(G)** Representative 3D confocal images of the distal ampulla in female filariae from WT, KO, and KO + DR hosts. *Wolbachia* are localized in the central rachis (arrow) and somatic sheath cells (lines) in parasites from WT and KO mice but are ether absent or present as patches in parasites from KO + DR mice. PH3 + nuclei are indicated by an asterisk (*).

### *Wolbachia* are dispensable for larval development in the arthropod vector but essential for L4 growth and maturation in the mammalian host

Filarial nematodes typically undergo 2 molts in the vector, followed by two other molts in the host accompanied by important growth and sexual maturation [8]. To clarify the potential *Wolbachia* requirement in these developmental processes in *L. sigmodontis,* we first analyzed the development of *Wb*(+) and *Wb*(-) microfilariae after ingestion by the arthropod vector. *Ornithonyssus bacoti* mites were fed on untreated and DR-treated microfilaremic *Il4rα-/-/Il5-/-* mice at 70 days post-infection (dpi). At that time, all circulating microfilariae from treated mice were *Wb*(–), while those from control mice consistently harbored *Wolbachia* (Fig 4A). Both *Wb*(+) and *Wb*(-) microfilariae successfully developed into infective L3 larvae over a two-week period (Fig 4B). FISH staining of L3 larvae confirmed that *Wb*(+) larvae retained *Wolbachia*, distributed along the antero-posterior axis, while *Wb*(-) larvae were completely devoid of *Wolbachia*. The similarity in L3 size (*Wb*(+): 0.727 ± 0.107 mm vs. *Wb*(-): 0.757 ± 0.048 mm, mean ± SD) and morphology between the two groups suggests that *Wolbachia* are not required for larval development or molting in the vector, as both types reached the infective L3 stage without significant differences.

Second, to assess the infective potential of *Wolbachia*-free L3 larvae, we inoculated *Wb*(+) or *Wb*(-) larvae into *Il4rα-/-/Il5-/-* mice to maximize parasite survival. We harvested the filariae at 14, 20, and 33 dpi to follow larval development post L3-to-L4 molt (14 dpi), through the rapid growth of L4 larvae (20 dpi) —when *Wolbachia* proliferate intensely and colonize the ovarian primordium in female filariae (S4 Fig and ref [41,45])— and after the transition into adulthood at 33 dpi (Fig 4A).

At 20 and 33 dpi, the number of recovered larvae from the pleural cavity and the parasite sex ratio were comparable between the two groups (Fig 4C and 4D), indicating that *Wolbachia* do not impact the larvae's ability to migrate to the pleural cavity or influence sex determination. Examination of the worms confirmed that larvae from both groups had undergone their third molt at 14 and 20 dpi, as indicated by the morphology of their buccal capsules [48] (Fig 4E). However, at 20 dpi, *Wb*(-) L4 larvae were significantly smaller than their *Wb*(+) counterparts (Fig 4F). By 33 dpi, all *Wb*(+) larvae had molted into adults, while all *Wb*(-) larvae remained in the L4 stage (68/68 *Wb*+ and 28/28 *Wb*-; Fig 4E). The size difference between the two groups became even more pronounced by this time, with *Wb*(-) larvae showing minimal growth after 20 dpi and no evidence of sexual dimorphism, in contrast to *Wb*(+) larvae, where females were longer than males (Fig 4F).

Examination of filariae at 33-dpi further illustrated these developmental differences (Fig 5). *Wb*(+) females had well-developed ovaries, with *Wolbachia* distributed along the germline (Fig 5A and 5B). By contrast, *Wb*(-) female worms showed impaired reproductive development, retaining only an ovarian primordium that did not progress beyond an early developmental stage (Fig 5C and 5D). In *Wb*(+) male worms, the testis extended along the entire length of the worm (arrows) and distinctive structures such as copulatory organs (the spicule) and sperm (small bright DAPI dots) were clearly visible, consistent with the development of fully differentiated reproductive tissues (Fig 5E and 5F). In contrast, testis development was compromised in *Wb*(-) males, no spermatozoa were visible and the spicule was absent (Fig 5E and 5G).

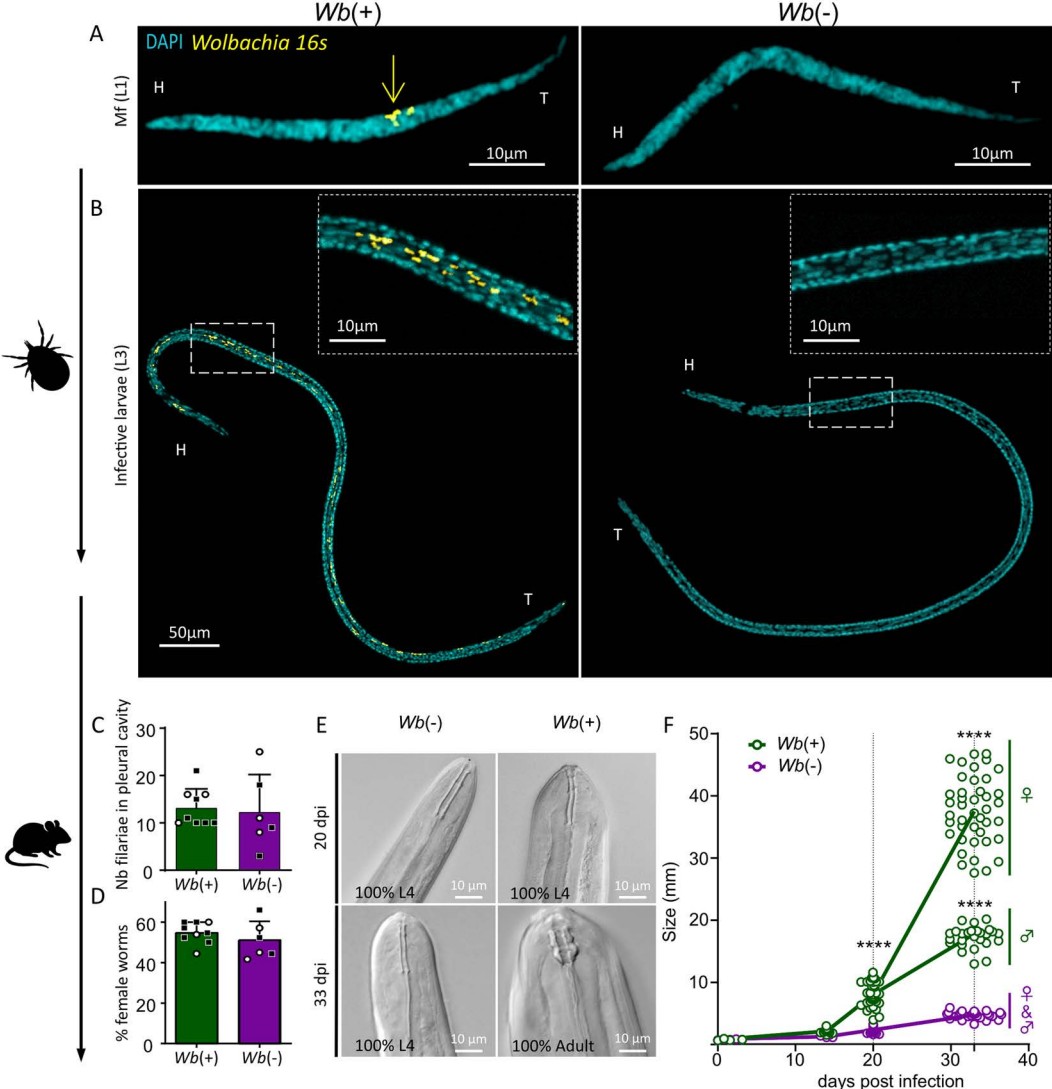

**Fig 4. _Wolbachia_-deficient microfilariae develop normally in the arthropod vector but exhibit delayed growth and differentiation after the third molt in the mammalian host. (A)** Representative confocal fluorescence images of microfilariae (L1) recovered from the blood of infected _Il4rα-/-/ Il5-/-_ mice at 70 dpi. _Wb_(+) and _Wb_(-) microfilariae were stained for DNA (DAPI, cyan) and _Wolbachia_ 16S ribosomal RNA (yellow). The yellow arrow highlights _Wolbachia_ clusters in _Wb_(+) worms, which are absent in _Wb_(-) worms. Head (H) and tail (T) are indicated to orientate the worms. **(B)** Representative confocal images of infective L3 larvae recovered from _Ornithonyssus bacoti_ mites fed on _Wb_(+) or _Wb_(-) microfilaremic _Il4rα-/-/Il5-/-_ mice. Insets show zooms on the squared areas, confirming the presence of _Wolbachia_ (yellow) in _Wb_(+) larvae and their absence in _Wb_(-) larvae. **(C-F)** To assess the developmental potential of _Wolbachia_-deficient L3 larvae, 40 _Wb_(+) or _Wb_(-) L3 larvae of _L. sigmodontis_ were inoculated into _Il-4rα-/-/Il-5-/-_ mice. Larvae were harvested at 14-, 20- or 33-days post-infection (dpi) for analysis. **(C)** Quantification of the number of _Wb_(+) and _Wb_(-) filariae recovered from the pleural cavity (PC) of _Il-4rα-/-/Il-5-/-_ mice at 20 (dots) and 33 (squares) dpi. **(D)** Proportion (%) of female filariae in the pleural cavity at 20 (dots) and 33 (squares) dpi. **(E)** Representative Differential Interference Contrast (DIC) images of the buccal capsule of worms at 20 and 33 dpi, allowing to differentiate L4 larvae and adult worms. L4 larvae have a buccal capsule composed of two thin walls while the capsule of adult worms displays three large segments. At 33 dpi, _Wb_(+) worms had fully molted into adults, while _Wb_(-) worms remained arrested in the L4 stage. 4-11 worms were analyzed at day 14, 32-39 at day 20 and 68 _Wb_(+) filariae and 28 _Wb_(-) filariae at 33dpi. **(F)** Growth dynamics of _Wb_(+) and _Wb_(-) filariae over time. Worm size (mm) is shown as median for each group, with individual worm data points displayed. n = 6-9 worms were measured at day 0, 4-11 at day 14, 32-39 at day 20. At day 33, 68 _Wb_(+) filariae and 28 _Wb_(-) filariae were analyzed. Statistical significance for size differences between groups was assessed using two-way ANOVA followed by Tukey's post-hoc test, **** p < 0.0001.

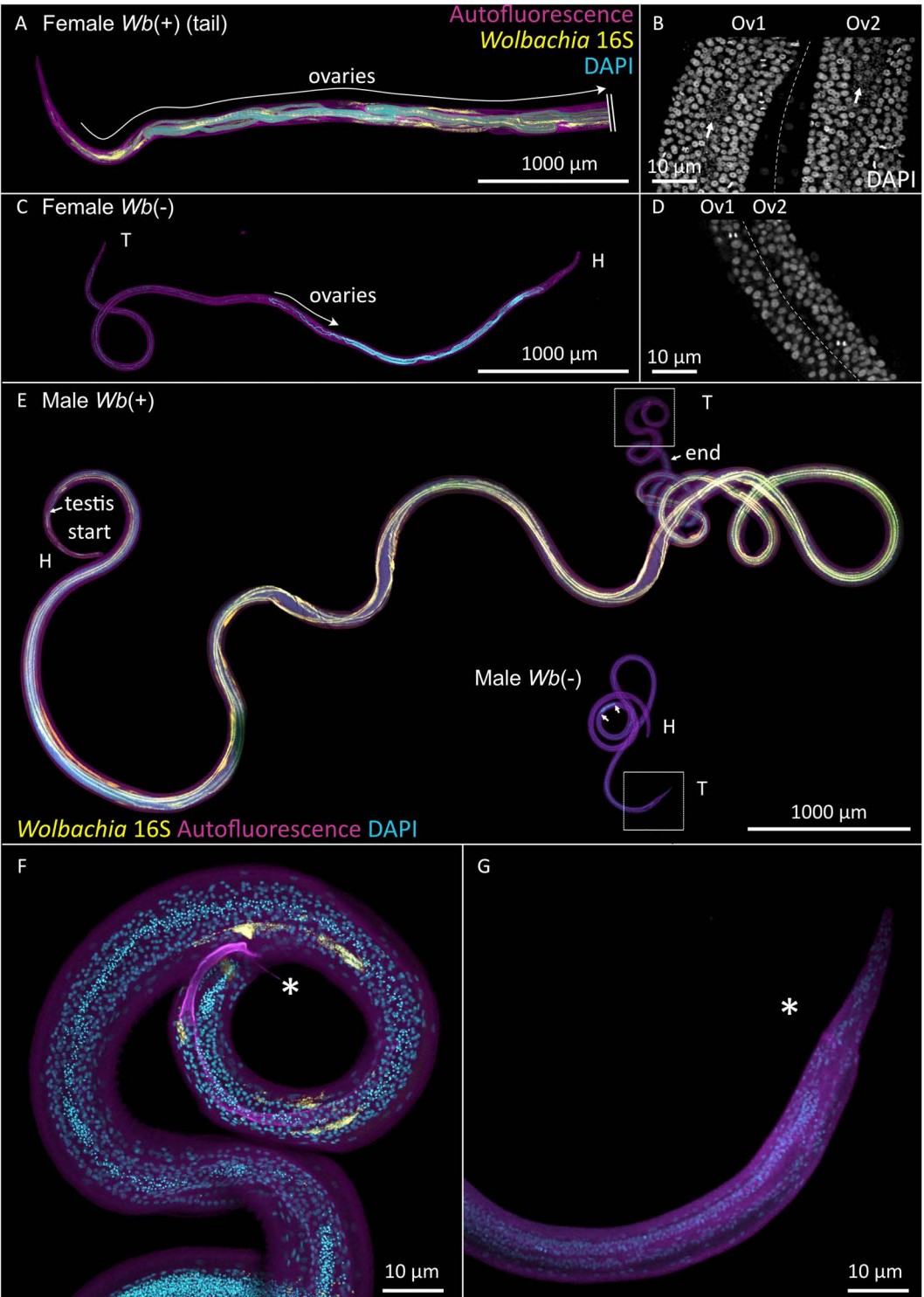

**Fig 5. Impact of *Wolbachia* on larval reproductive development.** Whole-mount confocal images of 33 dpi *Wb*(+) and *Wb*(-) *Litomosoides sigmodontis* worms stained for DNA (DAPI, cyan), *Wolbachia* (16S ribosomal subunit, yellow). Tissue autofluorescence is displayed in magenta. Head (H) and tail (T) are indicated to orientate the worms. **(A)** Tail of a *Wb*(+) female worm, cropped (indicated by\\) to fit the panel due to its length (total length ~4 cm). **(B)** High-magnification view of the proliferation zone of the *Wb*(+) ovaries. Note the dense organization of germline nuclei and abundant *Wolbachia* in the

rachis (arrow). **(C)** Whole *Wb*(-) female worm, with underdeveloped ovaries. **(D)** High-magnification view of the proliferation zone of the *Wb*(-) ovaries, showing the ovarian primordium devoid of *Wolbachia*. **(E)** Comparison of *Wb*(+) and *Wb*(-) male worms. The *Wb*(+) male exhibits a fully developed testicle running along the entire body length, terminating near the tail (arrows), while the *Wb*(delineated by 2 arrows) male shows an underdeveloped testicle. **(F)** Close-up of the tail of a *Wb*(+) male worm, showing a fully formed spicule (asterisk) and sperm (bright DAPI foci), indicative of mature reproductive tissues. **(G)** Close-up of the tail of a *Wb*(-) male worm, showing the absence of a spicule (arrow) and sperm.

These findings demonstrate that *Wolbachia* are not essential for filarial development in the arthropod vector. However, these endosymbionts become crucial for larvae in the mammalian host to support their growth, sexual differentiation and gonad development, otherwise arrested as abnormal L4 larvae.

## Discussion

Filarial parasites exhibit a complex lifecycle that alternates between an arthropod vector and a vertebrate host [8]. Successful development within the vector is essential for transmission, as the parasites must undergo two molts to progress from the microfilarial stage (L1) to infective larvae (L3). While *Wolbachia* are widely recognized as essential for the reproductive success of adult worms, their role during the early stages of the lifecycle, particularly in the vector, is still poorly understood.

Previous studies have suggested a critical role for *Wolbachia* in the early development of filarial nematodes, particularly during molts from L1 to L3 in *Brugia malayi*, *Onchocerca volvulus*, and *Litomosoides sigmodontis* [49–51] potentially through chitinase production facilitating exsheathment [49,52]. However, PCR analysis revealed incomplete *Wolbachia* depletion in these studies, which may have introduced competitive dynamics between *Wb*(+) and *Wb*(-) larvae, where residual *Wolbachia* would have conferred developmental advantages to *Wb*(+) larvae. Notably, *L. sigmodontis* microfilariae naturally harbor as few as 5–39 *Wolbachia* per worm [53], making full depletion difficult to confirm without spatially resolved techniques. By validating complete *Wolbachia* depletion using imaging techniques allowing the detection of single bacteria and introducing exclusively *Wb*(-) microfilariae into mite vectors, our study demonstrates that *L. sigmodontis* can successfully develop from L1 to L3 in the absence of *Wolbachia*. This challenges the assumed indispensability of *Wolbachia* during these early stages. However, as our study focused on *L. sigmodontis*, the possibility of species-specific roles for *Wolbachia* during early development in other filarial parasites cannot be excluded.

In the arthropod vector, L3 larvae remain quiescent until the blood meal, a phase characterized by metabolic dormancy and minimal growth similar to dauer larvae in *C. elegans* [54–56]. Upon inoculation into vertebrate hosts, *L. sigmodontis* L3 larvae resume growth as they migrate from the skin to the pleural cavity, where they molt into L4 larvae [8,57–59]. This phase is accompanied by significant *Wolbachia* proliferation [41], potentially influenced by host- or parasite-derived signals [56]. However, our study in type 2-deficient mice demonstrates that *Wb*(-) larvae successfully completed this migration and underwent the third molt, indicating that *Wolbachia* are dispensable for these processes in the absence of a competent immune response. These findings suggest that host-derived factors, such as temperature, hormonal signals, or metabolic cues, may be the main drivers of worm growth and molting processes at these stages in permissive environments. However, we cannot exclude the possibility that *Wolbachia* facilitate these processes indirectly. For example, *Wolbachia*-mediated activation of TLR2 has been shown to drive CCL17-dependent vascular permeability, which increases larval access to host tissues by enhancing mast cell degranulation and histamine release [60]. Understanding the interplay between host-derived signals and parasite development at these stages is crucial for elucidating the broader regulatory mechanisms underlying larval growth and adaptation.

Like other nematodes, parasitic species are highly responsive to environmental cues, which regulate key processes such as molting and reproduction. Two conserved receptor families—nuclear hormone receptors (NHRs) and insulin-like receptors (ILRs)—play central roles in integrating these signals [54,61]. In *C. elegans*, the NHR DAF-12 is activated by dafachronic acids, steroid-like molecules derived from cholesterol metabolism, and governs developmental progression

under favorable conditions [62,63]. In parasitic nematodes, DAF-12 has evolved to sense host-derived steroid hormones such as Δ4-dafachronic acid (Δ4-DA) [56,64], which can trigger the L3-to-L4 molt, as demonstrated in *Dirofilaria immitis* [56]. This adaptation highlights a metabolic dependency on host signals. Although the role of insulin signaling in filarial nematodes remains less understood, it may also contribute to synchronizing parasite development with host physiology [54,65,66]. Notably, type 2 immune responses are known to modulate host cholesterol and insulin signaling [67], suggesting that the immune environment may influence filarial development and *Wolbachia* maintenance indirectly, via these host-derived endocrine pathways.

While *Wolbachia* appear dispensable for L3 growth and the L3-to-L4 molt, its critical role becomes evident during the L4 stage. Wb(-) larvae showed severely impaired growth, failed to develop sexual dimorphism, and arrested at the L4 stage without progressing to adulthood, suggesting that host derived factors are not sufficient to perform these steps normally. These findings are consistent with prior studies on *Brugia pahangi* and *L. sigmodontis* [46,68], which demonstrated that early-stage *Wolbachia* depletion impairs larval development and prevents the molt to adulthood. In particular, Specht et al. [46] used suboptimal doxycycline protocols initiated shortly after L3 inoculation into the host and observed growth inhibition even when *Wolbachia* were only partially depleted. These results suggest that early disruption of *Wolbachia* function during the onset of exponential bacterial replication and tissue colonization can impair development prior to complete bacterial clearance. This suggests that a minimal bacterial density or activity is required to sustain metabolic signaling pathways necessary for L4 larvae maturation. While *Wolbachia* are critical for parasite development, their potential interactions with NHR and ILR pathways remain unexplored, highlighting the need for further research to unravel the connections between host endocrine signals, immune responses, and *Wolbachia*-mediated processes.

During the L4 stage, some *Wolbachia* migrate from the somatic lateral hypodermal chords to the gonadal primordium of female worms [41]. *Wolbachia* germline colonization ensures their maternal transmission, which is tightly linked with the reproductive success of the filarial host [12,14]. Wb(-) female worms, unable to establish this critical germline association, become reproductive dead-ends, highlighting the indispensable role of *Wolbachia* in the transmission of filarial nematodes. In males, where *Wolbachia* remain confined to the lateral chords and are absent from the testis, arrested development of Wb(-) male larvae suggests that the bacteria may exert systemic effects or signal indirectly to support male growth and sexual maturation. Additionally, *Wolbachia* may modulate immune defenses to protect larvae from host attack [14], enabling continued growth and development.

These findings align with strategies observed in insect hosts, where *Wolbachia* manipulate reproductive systems to favor its transmission, such as feminization and cytoplasmic incompatibility [12]. Similarly, in *L. sigmodontis*, absence of *Wolbachia* in larvae results in their inability to further develop into fertile adults, indirectly favoring the success of Wb(+) worms. This evolutionary mechanism emphasizes the deep interdependence between *Wolbachia* and their filarial hosts, ensuring the symbiotic relationship benefits both the parasite and the bacteria.

In adult *L. sigmodontis* worms as in most filarial species, *Wolbachia* occupy two distinct tissues: the somatic lateral hypodermal chords in both males and females, as well as the female gonads. Antibiotic-mediated depletion of *Wolbachia* leads to reproductive defects that worsen over time. It starts with germline defects while viable microfilariae are still being produced to eventually result in full sterility and aborted embryogenesis, leading to a dramatic reduction of circulating microfilariae. These findings are consistent with observations in *Brugia malayi*, where *Wolbachia* depletion similarly disrupts oocyte proliferation and proper development [13,20], ultimately impairing embryogenesis [44].

Our results also demonstrate that type 2 immune environment selectively reduce germline *Wolbachia* while sparing bacterial populations in the lateral hypodermal chords, suggesting that changes in the germline create conditions unfavorable for successful *Wolbachia* colonization or persistence. This selective impact highlights the importance of combining absolute quantification methods (like qPCR) with tissue-specific imaging to fully evaluate *Wolbachia* dynamics across different tissues, immune environments, time of infection and treatments. Despite this selective depletion in the germline, filariae in type 2-competent hosts exhibit reproductive impairments similar to those observed in worms from

type 2-deficient hosts treated with antibiotics. This highlights the indispensable role of germline *Wolbachia* in supporting reproductive fitness. While the direct role of somatic *Wolbachia* in lateral chords remains unclear, their persistence interrogates potential auxiliary functions, such as modulating host immune responses [15–17]. The selective nature of this immune driven effect suggests that the immune system does not directly target *Wolbachia*. Instead, the robust cuticle and surrounding tissues likely protect lateral chord and ovaries from cytotoxic granules, oxidative stress, and other immune effector mechanisms. This highlights the possibility that the immune environment indirectly influences germline *Wolbachia*. A non-permissive metabolic environment—driven by nutrient competition or disrupted endocrine signaling—could lead to ovarian defects, compromising the germline's ability to provide the necessary signals or metabolites to sustain *Wolbachia* proliferation. In turn, *Wolbachia* depletion may exacerbate this dysfunction by failing to support critical germline processes such as the germline genetic program [13]. This potential feedback loop highlights the delicate balance required to maintain germline and *Wolbachia* fitness in filarial nematodes, where disruptions to either component could have profound consequences for parasite reproduction and survival.

Understanding how the immune system shapes the balance that sustains parasite and *Wolbachia* fitness is crucial to elucidating these interactions. The pleural cavity's immune and metabolic landscape is influenced by the host genetic background during *L. sigmodontis* infection [25], potentially shaping both parasite fitness and *Wolbachia* persistence. Throughout the course of infection, millions of immune cells infiltrate the pleural cavity, with macrophages and eosinophils representing the dominant populations [27,28,69]. In resistant C57BL/6 mice, macrophages have been strongly associated with the ability to eliminate adult worms shortly after the final molt. However, the precise mechanisms underlying this resistance remain unknown. One potential factor is the efficient integration of monocytes into the long-lived resident macrophage (LCM) pool, a process that functions poorly in BALB/c WT mice and is absent in BALB/c KO mice lacking type 2 immune responses. LCMs in C57BL/6 mice exhibit an OXPHOS-dominant metabolic profile [38], consuming lipids and amino acids in a manner that could intensify competition for essential nutrients. By contrast, in BALB/c WT mice, monocyte-derived macrophages dominate the pleural cavity during infection [36–38]. These cells are metabolically distinct, relying less on OXPHOS metabolism [38], which may reduce lipid competition and facilitate cholesterol-dependent pathways such as DAF-12 signaling [61]. Additionally, arginine metabolism depends on type 2 responses. In permissive environments like BALB/c *Il4r*[-/-]/*Il5*[-/-] mice, reduced iNOS activity might allow arginine to remain accessible [36], potentially supporting parasite growth and *Wolbachia* survival. This altered metabolism may also induce competition for other essential amino acids, further influencing parasite development. Altogether, these changes may create a permissive environment that facilitates both parasite development and *Wolbachia* maintenance. Whether these effects result from direct nutrient availability, reduced metabolic competition, or active sensing of host-derived signals remains to be determined.

Leveraging the immune and metabolic pathways implied in *Wolbachia* and filarial fitness could enhance the effectiveness of *Wolbachia*-targeted therapies by creating a hostile environment that limits bacterial survival and parasite fitness. This is supported by our recent findings [70] and suggests that combining immunostimulatory compounds with anthelmintics could enhance the effectiveness of anti-filarial treatments by boosting the host's immune response. However, balancing type 2 responses is essential to avoid immune-driven pathology. While enhancing type 2 responses could effectively reduce *Wolbachia* load and impair filarial reproduction, there is a risk of exacerbating immune-mediated tissue damage [33,36,71]. Combinations of anti-*Wolbachia* compounds and benzimidazoles such as albendazole may overcome this risk. Albendazole is a direct-acting anthelmintic that was shown to provide synergistic effects [72,73] with anti-*Wolbachia* compounds, but the mechanism is not completely understood yet and requires further research. A better understanding of parasite-specific molecular pathways could further improve integrated therapeutic approaches. For instance, the fact that NHR-8 has been implicated in ivermectin resistance [74] and the regulation of parasite development by DAF-12 [56,75] both highlight these pathway as promising therapeutic targets in parasitic nematodes. These examples underscore how targeting metabolic and developmental pathways could complement *Wolbachia*-targeted strategies, providing additional avenues to address challenges such as drug resistance and incomplete parasite clearance.

## Conclusion

Our study highlights the multifaceted roles of *Wolbachia* in filarial nematode development and reproduction, revealing their critical contributions to female germline function, with a clear developmental threshold at the L4 stage (Fig 6). By uncovering the intertwined relationship between female germline development and *Wolbachia* proliferation, we demonstrate a mutual dependency: a favorable germline environment supports proper *Wolbachia* proliferation, which in turn ensures proper germline development and reproductive success. This delicate balance underscores the complex interplay between host immune environment, parasite biology, and *Wolbachia* survival, offering new insights into the factors shaping parasite fitness and reproductive strategies. These findings not only provide a foundation for advanced therapeutic approaches but also deepen our understanding of host-parasite-endosymbiont interactions.

## Methods

### Ethics statement

All experimental procedures were conducted in strict accordance with EU Directive 2010/63/EU and relevant French national legislation (Décret No. 2013–118, 1er février 2013, Ministère de l'Agriculture, de l'Agroalimentaire et de la Forêt). Protocols were approved by the ethical committee of the Museum National d'Histoire Naturelle (CEEA 68 Cuvier, Project agreements #13838 and #16091) and by the Direction départementale de la cohésion sociale et de la protection des populations (DDCSPP, No. D75-05–15).

### Mice, infestation, and treatments

*Il-4rα⁻ᐟ⁻/Il-5⁻ᐟ⁻* [33–36,76] (University Hospital Bonn, Germany) and control female BALB/c mice (Envigo) were maintained and bred in the MNHN facilities, with a 12-hour light/dark cycle and access to food and water ad libitum.

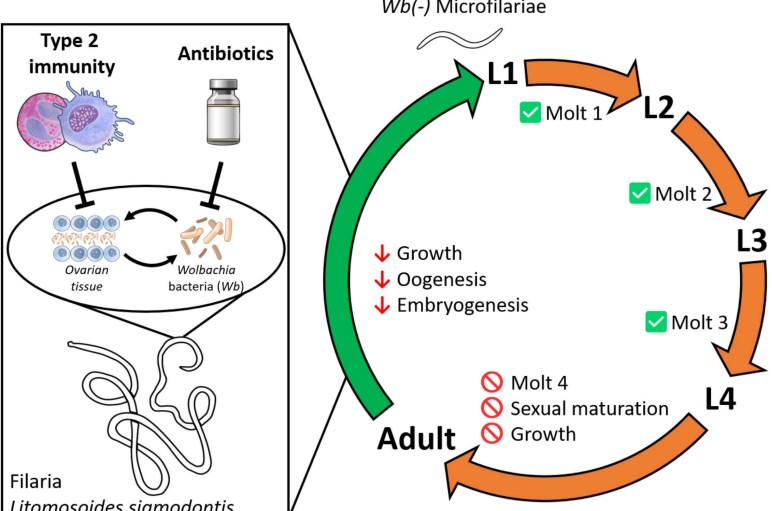

**Fig 6. The key roles of *Wolbachia* on filarial biology.** In adult female *Litomosoides sigmodontis, Wolbachia* and the germline maintain a mutual dependency: *Wolbachia* are required to support oogenesis, while a functional germline environment sustains Wolbachia proliferation. Disruption of this balance—either by type 2 immune environment affecting the ovarian tissue or by direct antibiotic depletion of *Wolbachia*—initiates a cascade of defects. These include reduced *Wolbachia* density in the germline, defective oogenesis, stunted parasite growth, and eventually, reproductive failure. Despite these impairments, adult females can still transiently release microfilariae lacking *Wolbachia*. These *Wolbachia*-depleted microfilariae can develop into infective L3 larvae within the mite vector. However, once transmitted to the vertebrate host, they arrest at the L4 stage, failing to grow, molt, or differentiate into adults. This indicates the existence of a developmental threshold at which *Wolbachia* becomes essential for sustaining growth, sexual maturation, and successful progression to adulthood. Illustrations adapted from NIH BioArt Source and CDC PHIL (Public Domain).

Maintenance of the filaria *L. sigmodontis* Chandler, 1931 and isolation of infective larvae (L3) from the mite vector, *Ornithonyssus bacoti*, were carried out as previously described [25,36,77]. Briefly, one-week-starved mites were allowed to blood-feed on infected *Meriones unguiculatus* jirds (700–2000 microfilariae/mm³ in the peripheral blood) overnight. Then blood-fed mites were recovered and kept at 28°C and the 80% relative humidity to allow *L. sigmodontis* development from microfilariae into L3. L3 larvae were recovered by dissecting mites in warm RPMI medium under a stereomicroscope and used immediately for mouse infection [25,26,33,36,77]. Similarly, mites were fed overnight on control or antibiotic-treated microfilaremic *Il4rα⁻/⁻/Il5⁻/⁻* mice to analyze the development of *Wolbachia*-negative microfilariae. After 14 days, the mites were dissected to recover the larvae.

All experiments were performed using 6- to 8-weeks old female mice to ensure higher worm recovery and microfilaremia than in males [78]. Mice were infected subcutaneously in the neck with 40 infective L3 larvae of *L. sigmodontis* in 200 µL of RPMI medium (Eurobio). Infections were allowed to progress for various time points depending on the experimental design.

An antibiotic treatment was optimized from previous studies on mice [46,47] to analyze the role of *Wolbachia* in filarial development and fertility in *Il4rα⁻/⁻/Il5⁻/⁻* mice without affecting filarial recovery (J. Gal, PhD thesis, MNHN, 2023). Mice received doxycycline (50 mg/kg bid; Sigma-Aldrich, D3447) and rifampin (5 mg/kg bid; Sigma-Aldrich, R3501), starting at 40 days post-infection (dpi). Doxycycline treatment continued for 14 days, while rifampin was administered for the first 5 days. Post-treatment, mice were monitored until 70 dpi for parasite collection and analysis.

### Isolation and measure of filariae

Mice were sacrificed at different times post-inoculation (dpi) from 8 to 70 dpi. Filariae were collected by flushing the pleural cavity 10 times with 1ml cold phosphate buffered saline (PBS) as previously described [33,58,79]. After counting and sexing, filariae were fixed in 4% paraformaldehyde (PFA) or flash-frozen in liquid nitrogen and stored at -80°C for subsequent analyses. For worm measurements, filariae were imaged at low resolution on a stereo microscope (AxioZoom V16, Zeiss) and measured with the polyline tool using Zen Blue software. Larval stages were evaluated by investigating the buccal capsule by Differential Interference Contrast (DIC) imaging on a BX62 microscope (Olympus) equipped with a 40X oil-immersion objective.

### Determination of microfilaremia

Peripheral microfilariae were quantified using a 10 µL thick blood smear prepared from retro-orbital blood collection using a 10 µL capillary tube. The smear was stained with Giemsa, and the stained slides were then scanned using a NanoZoomer slide scanner (Hamamatsu). Digital images were visually inspected and analyzed using QuPath software version 0.5.0 [80]. A pixel classification algorithm was trained and applied to accurately detect and quantify individual microfilariae within the smear, while excluding leukocytes, debris, and other artifacts. A size filter was applied to ensure that only objects within the expected size range for microfilariae were counted.

### Quantification of *Wolbachia* by quantitative PCR

Female *L. sigmodontis* filariae, previously isolated and stored at -80°C, were used for DNA extraction. Each filaria was placed in a 2 mL microtube containing 100 µL of lysis buffer (Buffer ATL, Qiagen) and three 5 mm tungsten beads. The samples were homogenized using a TissueLyser II (Qiagen) with two cycles of 1 minute at 20 Hz. Following a brief centrifugation, an additional 80 µL of lysis buffer was added to rinse the beads. The entire lysate was transferred to a new tube, and 20 µL of proteinase K (Qiagen) was added. The mixture was vortexed briefly and incubated overnight at 56°C with regular agitation. DNA was then purified using the Qiagen DNA MicroKit according to the manufacturer's instructions. Finally, DNA was eluted with 50 µL of DNAse-free water and stored at -20°C. The quantity of extracted DNA was

measured using the Qubit dsDNA High Sensitivity Kit (Invitrogen). Samples were diluted to a concentration of 0.25 ng/µL to avoid bias due to varying concentrations among samples. The quality of the DNA was verified using spectrophotometry (Nanodrop 2000, Thermo Scientific). Primers were designed based on established protocols [36]. For *Actin* (Ls-*Actine*), the sequences were: Forward: GGCCGAACGTGAAATTGTACGTG, Reverse: GACCATCGGGCAATTCATACGACT. For *Wolbachia ftsZ* (wLs-*ftsZ*), the sequences were: Forward: TACGGCGCACACCTTCAAAGT, Reverse: CTTTTCATTACGGCAGGGATGGGT.

Quantitative PCR (qPCR) was performed to assess the *Wolbachia* bacterial load in female filariae at 24-, 28-, 30-, 32-, 50-, 60-, and 70-days post-infection (dpi). qPCR was conducted on a LightCycler 480 thermocycler (Roche Diagnostics, France) using the SensiFAST SYBR No-ROX Kit (Bioline). The reaction mixture for each sample consisted of 5 µL of MasterMix, 1 µL of forward and reverse primers (5 µM each), and 4 µL of cDNA at 0.25 ng/µL (total 1 ng). Each sample was analyzed in triplicate.

The amplification program included an enzyme activation step at 95°C for 2 minutes, followed by 45 cycles of denaturation at 95°C for 10 seconds, primer annealing at 60°C for 5 seconds, and elongation at 72°C for 10 seconds. A melting curve analysis was performed at 65°C for 1 minute, followed by a gradual increase to 95°C with continuous data acquisition to control for specific amplification.

The expression levels of the *Wolbachia ftsZ* gene and the *L. sigmodontis* β-*actin* gene were assessed using absolute quantification against a plasmid standard curve containing known concentrations of the *ftsZ* or β-*actin* gene [81]. The initial *Wolbachia* load was expressed as the ratio (R) of *Wolbachia ftsZ* gene copies to *L. sigmodontis actin* gene copies, calculated as R = wLs-*ftsZ*/Ls-act. To account for worm size differences and ensure meaningful comparisons, the absolute density of *ftsZ* and *actin* gene copies was calculated. This was achieved by multiplying the number of gene copies per nanogram of DNA (as determined by qPCR) by the total quantity of DNA extracted from each sample, providing the total number of gene copies per worm. This total was then normalized by the average worm length in the group, resulting in a final expression of gene copy density as "number of gene copies per millimeter" (copies/mm). This dual approach of ratio and absolute density measurements allowed for a comprehensive analysis of *Wolbachia* load in relation to both bacterial and host genomic content, as well as the physical size of the worms.

### RNA fluorescence *In Situ* Hybridization (FISH) and phospho-histone 3 (PH3) staining

Frozen *L. sigmodontis* female worms were thawed and dissected in cold PBST (PBS 1X with 0.1% Tween 20). Whole adult worms, dissected ovaries, or larvae were carefully isolated and fixed in 3.6% paraformaldehyde (PFA) for 20 minutes for whole worms, or 10–15 minutes for dissected ovaries, L3 and L4 larvae. The samples were then rinsed three times with PBST. Following dissection and fixation, PBST was replaced with 100 µL of hybridization buffer containing FISH probes for whole worms or 50 µL for isolated ovaries. The hybridization buffer consisted of 500 µL deionized formamide, 200 µL 20X SSC, 10 µL 50X Denhardt's solution, 100 µL 1M DTT, 10 µL 10% Tween 20, and 180 µL 50% Dextran Sulfate buffer (prepared in advance and stored at -20°C). The probes were used at a final concentration of 0.5 µM, targeting *Wolbachia* 16S rRNA, as previously described [82]. The following oligonucleotides were used: W1: [CY3]CCCCAGGCGGAATGTTTA, W2: [CY3]CTTCTGTGAGTACCGTCATTATC, W1 H1: ACGCGTTAGCTGTAATAC, W1 H2: TAATCTTGCGATCGTAGT, W2 H1: TTCCTCACTGAAAGAGCTTT, and W2 H2: CACGGAGTTAGCCAGGACT. Samples were incubated at 37°C for a minimum of 16 hours in the dark to prevent photobleaching of the fluorescent probes.

Following hybridization, the hybridization buffer was removed, and the samples were rinsed with 500 µL of 1X SSCDT (SSC with 0.1% Tween 20, 1 mM DTT) at room temperature. The samples were then incubated in fresh 1X SSCDT for 1 hour at 42°C, followed by a wash with 0.5X SSCDT at room temperature. A final incubation with 0.5X SSCDT was performed for 1 hour at 42°C. After washing with PBST, samples were stored at 4°C until further processing.

Primary antibodies against PH3 (Invitrogen, clone 9H12L10, rabbit monoclonal) were diluted 1:500 in antibody diluent (PBS supplemented with 10% fetal bovine serum [FBS], 1% bovine serum albumin [BSA], 0.1% Triton X-100, and 0.05% sodium azide), and 50 µL of the solution was added per ovary. Samples were incubated overnight at 4°C. Following the primary

antibody incubation, samples were rinsed once with PBST, followed by a 10-minute wash with PBST. Cy5-conjugated Goat anti-Rabbit secondary antibodies (Invitrogen, polyclonal, diluted 1:500 in antibody diluent) and DAPI (1:500) were added to the samples for a 6-hour incubation at 4°C. Samples were then rinsed once with PBST and washed again for 10 minutes in PBST. Prepared samples were mounted on poly-lysine slides using a drop of PBS. The ovaries were gently placed on the slide and stretched by holding the proximal end. After drying, a few drops of mounting medium (Prolong Diamond, Invitrogen) were added, and the slides were covered with 50 mm coverslips. The mounted slides were stored at 4°C before imaging.

The larvae, the proximal uteri and the proliferative zone of the ovaries of adult filariae were imaged on a Zeiss LSM880 confocal microscope. Acquisition was performed using the 32-channel Gallium arsenide phosphide (GaAsP) spectral detector with a 20×objective as previously described [36,71,83]. Samples were excited simultaneously with 405, 561, and 633 nm laser lines, and signals were collected using the linear array of 32 GaAsP detectors in lambda mode, with a spectral resolution of 8.9 nm across the visible spectrum. Spectral images were then unmixed using Zen software (Carl Zeiss), using reference spectra acquired from slides labeled with single fluorophores. When necessary, images were stitched, and maximum intensity projections were generated prior to image analysis.

### Image analysis

Samples were imaged using a Zeiss LSM880 confocal microscope as described above. Quantification of *Wolbachia* in the lateral chords was performed by segmenting the lateral chords in 3D using Imaris software v9.9 (Bitplane) with the surface tool. The segmented volume of *Wolbachia* within the lateral chords was then normalized to 100 µm of worm length, allowing for consistent and accurate comparisons of *Wolbachia* density across samples.

For the analysis of the ovaries, the gonad was imaged with a z-stack capturing the full depth of the tissue. The digital images were processed and ImageJ macro scripts were utilized as previously described [13] to semi-automatically quantify the number and distribution of cells within the ovarian proliferative zone (PZ). The gonad was first digitally linearized using the straightening function in ImageJ. The linearized images were then segmented into 100 µm-wide sections starting from the distal tip of the ovary. For each section, the total number of mitotic cells was quantified by automatically detecting PH3+foci, the area occupied by *Wolbachia* was quantified within the same sections by thresholding the *Wolbachia* 16S rRNA signal. The density of mitotic nuclei and *Wolbachia* within the PZ were analyzed by summing values obtained in all the sections. The accuracy of the automated analysis was validated by comparing the results with manual counts on a subset of samples. Pyknotic nuclei were manually counted in the ovarian images. These were identified as bright DAPI foci that were negative for PH3 staining, indicating apoptotic cells.

The analysis of the content of the proximal uteri was performed on single slices from confocal z-stacks using QuPath software version 0.5.0 [80]. A pixel classification algorithm was applied to separate microfilariae from aborted embryos and background artifacts. However, due to the high density of the microfilariae, the algorithm failed to separate individual objects and detected these as large contiguous areas containing all the microfilariae or embryos within the section. To estimate the number of microfilariae and embryos in these annotations, the total area of each detected region was therefore divided by the average area of a single microfilaria or embryo. This provided an estimated count of microfilariae and embryos within each section. To account for variations in worm orientation during imaging, the results were normalized to a standardized section length of 100 µm, ensuring consistent comparisons across samples. The "positive cell detection" tool was then used to identify and count *Wolbachia* foci within the microfilariae, allowing for the estimation of the percentage of *Wolbachia*-positive microfilariae (100 * number of *Wolbachia* foci/ estimated number of microfilariae).

### Statistical analyses

All statistical analyses were performed using GraphPad Prism version 10.0 (GraphPad Software, San Diego, CA, USA). Data were analyzed using appropriate statistical tests based on normality (Shapiro-Wilk test) and homogeneity of variances (Bartlett's test).

For comparisons involving two independent groups, unpaired t-tests were used for normally distributed data, and the Mann-Whitney U test was employed for non-normally distributed data.

For comparisons involving more than two groups, one-way ANOVAs were conducted for normally distributed data, followed by Tukey's multiple comparisons test to assess specific differences between groups. For non-normally distributed data, the Kruskal-Wallis test was used, followed by Dunn's post-hoc test.

Two-way ANOVAs were conducted to evaluate differences across time points and between Type 2-deficient and wild-type groups, followed by Bonferroni's multiple comparisons test to assess specific differences. Differences between time points were tested independently within each group, and between groups at the same time point.

Quantitative data are presented as mean ± standard error of the mean (SEM) unless otherwise stated. For non-parametric data, medians and interquartile ranges or standard deviation (SD) are reported. Statistical significance was set at $p < 0.05$. The specific statistical tests applied to each experiment are detailed in the corresponding Figure legends.

All graphs were generated using GraphPad Prism, and results were visualized with appropriate error bars to indicate the variability of the data. Significance levels are indicated as follows: *$p < 0.05$, **$p < 0.01$, ***$p < 0.001$, and ****$p < 0.0001$.

## Supporting information

**S1 Fig. Growth and actin gene expression of female *Litomosoides sigmodontis* in wild-type and type 2-deficient mice.** Type 2-competent wild-type (WT) and type 2-deficient (*Il4rα⁻/⁻/Il5⁻/⁻*, KO) mice were inoculated with 40 infective larvae (L3) of the filaria *L. sigmodontis*. Parasites were harvested and measured at various time points before and after the fourth molt (~30 dpi), and *Wolbachia's* gene *ftsZ* and filarial *actin* were evaluated by qPCR in female filariae. (A) Measurements of worm length in millimeters (mm) from 24 to 70 days post-infection (dpi) in wild-type and type 2-deficient (*Il4rα⁻/⁻/Il5⁻/⁻*) mice. Results are expressed as the mean ± SD of n = 20–40 filariae per group (24–50 dpi), and n = 8 filariae from wild-type and 28 filariae from *Il4rα⁻/⁻/Il5⁻/⁻* hosts (60 dpi). Two-way ANOVAs followed by Bonferroni's multiple comparisons tests were performed; ***p < 0.001, ****p < 0.0001 indicate significant difference between filariae from wild-type and *Il4rα⁻/⁻/Il5⁻/⁻* hosts. (B) Quantification of *actin* gene expression, normalized to worm length, over the same period. These measurements provide a baseline for the relative quantification of *Wolbachia* density shown in Fig 1, accounting for changes in worm size that could influence the interpretation of bacterial load. Results are expressed as the mean ± SD of n = 4–6 filariae per group (24–60 dpi), n = 11–15 filariae per group (70 dpi). Two-way ANOVAs followed by Bonferroni's multiple comparisons tests were performed; **p < 0.01, ***p < 0.001 indicate significant difference between filariae from wild-type and *Il4rα⁻/⁻/Il5⁻/⁻* hosts. (PDF)

**S2 Fig. Effect of host immune background and *Wolbachia* depletion on filarial size and pleural cavity burden at 70 dpi.** (A) Number of filariae recovered in the pleural cavity of WT, KO, and KO + DR mice at 70 dpi. Results are expressed as mean ± SEM, with individual data points representing each mouse (n = 6–8 mice per group). One-way ANOVA followed by Tukey's multiple comparison test were performed. **p < 0.01, ns: not significant. (B) Female filariae size (cm) from WT, KO, and KO + DR mice at 70 dpi. Results are expressed as mean ± SEM, with individual data points representing each measured worm (n = 6–8 worms per group). One-way ANOVA followed by Tukey's multiple comparison test were performed. **p < 0.01, ***p < 0.001. (PDF)

**S3 Fig. Anatomical organization of the female reproductive system and *Wolbachia* localization in *Litomosoides sigmodontis.*** (A) Fluorescent microscopy images of an adult female filaria highlighting key anatomical regions. The distal ampulla and ovaries are situated at the posterior end, transitioning through the oviduct into the uteri, which extend along the body axis. The proximal uteri, near to the ovojector, contain mature microfilariae.

Staining highlights nuclei (DAPI, blue) and actin filaments (gray). (B) Schematic representation of the female reproductive system. The ovary includes a proliferative zone (PZ, red), where germ cells divide, and a meiotic zone (MZ, blue), where meiosis occurs. The uteri (orange) extend anteriorly and terminate at the ovojector, where mature microfilariae are expelled. (C) Diagram of the distal ovary, showing the rachis (central cytoplasmic core) surrounded by germ cells. *Wolbachia* (yellow) are distributed along the rachis and within some sheath cells. The ampulla marks the distal tip of the ovary and the starting point of oogenesis. In somatic tissues, *Wolbachia* also reside in the lateral hypodermal chords (see Fig 2E).
(PDF)

**S4 Fig. Colonization of the ovarian primordia by *Wolbachia* in L4 female filariae.** (A) Quantification of *Wolbachia* density ($\mu m^2$/mm) in *Litomosoides sigmodontis Wb*(+) larvae at different days post-inoculation, based on fluorescence microscopy images of entire larvae. Brown-Forsythe ANOVA test followed by a Dunnett's T3 multiple comparisons post-hoc test (***$p < 0.001$). (B) Confocal fluorescence image of a *Wb*(+) L4 female *L. sigmodontis* at day 14 post-infection (D14). The worm was stained for DNA (DAPI, cyan) and *Wolbachia* (16S ribosomal RNA, yellow). The anterior-to-posterior orientation is indicated (h = head; t = tail). (C) Magnified region showing the lateral hypodermal chords and ovarian primordia in the boxed area from (B). *Wolbachia* are visible in the lateral chords (Chord1 and Chord2) and are invading the ovarian primordia.
(PDF)

**S1 Table. Raw data of the graphs.**
(XLSX)

## Acknowledgments

We would like to thank Cyril Willig and Xavier Marques from the Plateforme de microscopie photonique et imagerie du Muséum (CeMIM), as well as the Plateau technique de Cytométrie en flux et QPCR (QCYT) of the Plateforme analytique du MNHN (PAM) for their support. We also acknowledge the imaging facility MRI, member of the national infrastructure France-BioImaging (https://ror.org/01y7vt929) supported by the French National Research Agency (ANR-24-INBS-0005 FBI BIOGEN).

**Declaration of Generative AI in the writing process:** During the preparation of this work, FF used ChatGPT-4 (OpenAI, January 2024 version) to improve the language and readability of the manuscript, as well as to perform quality checks, such as ensuring concordance between Figure legends and results. After using this tool, the authors reviewed and edited the content as needed and take full responsibility for the content of the publication.

## Author contributions

**Conceptualization:** Frédéric Fercoq, Estelle Remion, Frédéric Landmann, Coralie Martin.

**Formal analysis:** Frédéric Fercoq, Estelle Remion.

**Funding acquisition:** Frédéric Fercoq, Coralie Martin.

**Investigation:** Frédéric Fercoq, Clément Cormerais, Estelle Remion, Joséphine Gal, Julien Plisson, Arame Fall.

**Resources:** Marc P. Hübner, Frédéric Landmann, Coralie Martin.

**Visualization:** Frédéric Fercoq, Clément Cormerais, Estelle Remion.

**Writing – original draft:** Frédéric Fercoq, Frédéric Landmann, Coralie Martin.

**Writing – review & editing:** Frédéric Fercoq, Estelle Remion, Joséphine Gal, Julien Plisson, Arame Fall, Joy Alonso, Nathaly Lhermitte-Vallarino, Marc P. Hübner, Linda Kohl, Frédéric Landmann, Coralie Martin.

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
