## [Decision Letter · Decision Letter 0]

PPATHOGENS-D-25-00409

Host environment shapes filarial parasite fitness and Wolbachia endosymbionts dynamics

PLOS Pathogens

Dear Dr. Martin,

Thank you for submitting your manuscript to PLOS Pathogens. After careful consideration, we feel that it has merit but does not fully meet PLOS Pathogens's publication criteria as it currently stands. Therefore, we invite you to submit a revised version of the manuscript that addresses the points raised during the review process.

Please submit your revised manuscript within 30 days Jul 01 2025 11:59PM. If you will need more time than this to complete your revisions, please reply to this message or contact the journal office at plospathogens@plos.org. Please include the following items when submitting your revised manuscript:

We look forward to receiving your revised manuscript.

Kind regards,

Keke C. Fairfax, PhD

Academic Editor

PLOS Pathogens

Jeffrey Dvorin

Section Editor

PLOS Pathogens

Sumita Bhaduri-McIntosh

Editor-in-Chief

PLOS Pathogens

orcid.org/0000-0003-2946-9497

Michael Malim

Editor-in-Chief

PLOS Pathogens

orcid.org/0000-0002-7699-2064

**Journal Requirements:**

1) We do not publish any copyright or trademark symbols that usually accompany proprietary names, eg ©,  ®, or TM  (e.g. next to drug or reagent names). Therefore please remove all instances of trademark/copyright symbols throughout the text, including:

- ® on page: 35.

3) Some material included in your submission may be copyrighted. According to PLOSu2019s copyright policy, authors who use figures or other material (e.g., graphics, clipart, maps) from another author or copyright holder must demonstrate or obtain permission to publish this material under the Creative Commons Attribution 4.0 International (CC BY 4.0) License used by PLOS journals. Please closely review the details of PLOSu2019s copyright requirements here: PLOS Licenses and Copyright. If you need to request permissions from a copyright holder, you may use PLOS's Copyright Content Permission form.

Potential Copyright Issues:

- Figures 4 and 6. Please confirm whether you drew the images / clip-art within the figure panels by hand. If you did not draw the images, please provide (a) a link to the source of the images or icons and their license / terms of use; or (b) written permission from the copyright holder to publish the images or icons under our CC BY 4.0 license. Alternatively, you may replace the images with open source alternatives. See these open source resources you may use to replace images / clip-art:

4) We note that your Data Availability Statement is currently as follows: "All data generated or analyzed during this study are included in this published article and its supplementary information files. Further inquiries can be directed to the corresponding author.". Please confirm at this time whether or not your submission contains all raw data required to replicate the results of your study. Authors must share the “minimal data set” for their submission. PLOS defines the minimal data set to consist of the data required to replicate all study findings reported in the article, as well as related metadata and methods (https://journals.plos.org/plosone/s/data-availability#loc-minimal-data-set-definition).

- The points extracted from images for analysis..

**Reviewers' Comments:**

Reviewer's Responses to Questions

**Part I - Summary**

Reviewer #1: This is a very interesting study that provides important insights into the role of the host immune system in shaping Wolbachia dynamics and filarial parasite fitness. Notably, this study provides new insights into the particular susceptibility of the reproductive tissues of filarial nematodes to type 2 immunity, whereby type 2 immunity selectively degrades the development of oocysts, with downstream impact on Wolbachia colonisation of those cells. Wolbachia-negative microfilariae exhibited apparently normal larval development including in the vector, but failed to reach sexual maturity in the definitive host. This is an important step in identifying the target of protective immunity against filarial nematodes, and in clarifying the causal relationships between type 2 immunity, Wolbachia dynamics, and filarial parasite fitness. The experiments are elegantly designed and the results are well presented, with excellent use of RNA fluorescence in situ hybridisation and histology.

Reviewer #2: The manuscript by Fercoq et al. presents a relevant contribution to the understanding of the interaction between the host immune system and the Wolbachia-filariae symbiosis, revealing mechanisms that directly impact the life cycle and reproductive capacity of the parasites. The results obtained provide important support for the development of more effective therapeutic strategies to combat filariasis, reinforcing the potential of the immune system as a modulator of the efficacy of anti-Wolbachia therapies.

**Part II – Major Issues: Key Experiments Required for Acceptance**

Reviewer #1: (No Response)

Reviewer #2: I have no suggestions for major revisions.

**Part III – Minor Issues: Editorial and Data Presentation Modifications**

Reviewer #1: I have a few comments and suggestions that I believe would improve the clarity of the manuscript.

- In the introduction, it would be good to mention whether / how type 1 and Th17 immunity might affect filarial survival and Wolbachia dynamics. This would help to clarify the novelty of the study, and the importance of the type 2 immune response in this context.

- Although FISH images don't show any Wolbachia post DR treatment, Fig 2A suggests that DR treatment in KO mice does not completely remove Wolbachia (these figures show normalised values (x 1,000) relative to average worm size). Could these residual levels explain the ability of apparently Wolbachia-negative microfilariae and larvae to develop somewhat normally? For example, as per the statements lines 209-210 & 507-509, those low numbers of Wolbachia might still provide some benefit, which would wane as the larvae increase in size at L4 and adult stages while Wolbachia are unable to proliferate due to early “sterilisation” by DR. It would be good to clarify this point in the discussion, and if so, to temper the wording in lines 456-458.

- In the discussion, paragraph lines 479-500 seems very speculative, and could be shortened.

- Figure 6: shouldn't the arrow linking type 2 immunity (top left) to Wolbachia instead point to the worm? I wonder if here the authors might be able to graphically differentiate between the direct effects of antibiotics and type 2 immunity on Wolbachia and on the filaria's development, respectively, since it appears that the effects of immunity on Wolbachia are indirect (through the worm). This could be clarified in the figure legend.

Reviewer #2: Minor Revision

1.In the introduction section

a) Standardizing technical terms is essential. For example, when referring to the “host immune environment” or the “type 2 immune response”, consistency should be maintained throughout the text.

• "type 2 immune deficient mice", "BALB/c mice strains", "immune-competent mouse strains" → can be standardized to avoid unnecessary variations.

b). Before describing the worm's anatomy, a sentence like:

"To better understand how Wolbachia interact with their nematode hosts at the tissue level, we briefly describe the anatomy relevant to their localization."would greatly help the reader's understanding.

2. Discussion section

A) after describing the possible metabolic competition in the permissive environment (lines 573–581), there is no clear conclusion or testable hypothesis. A more explicit conclusion would help with understanding.

b) since the discussion addresses multiple immunological, metabolic pathways and anatomical sites, a schematic figure would help the reader understand the complexity of the interactions.

3. Conclusion section

The sentence "revealing their critical contributions to germline function and its stage-specific dependencies" is somewhat generic in my understanding. I understood that the great contribution of the work is precisely to show that the dependency is crucial from the L4 stage onwards.

4. Methods sextion

On line 633-635 include some detail on how the life cycle of L. sigmodontis was maintained in the laboratory. Since the infection was in females, include a comment on how the infection or immune response is more consistent or reproducible in females when compared to males. Also include how the negative control was performed in this experiment.

PLOS authors have the option to publish the peer review history of their article (what does this mean? ). If published, this will include your full peer review and any attached files.

**Do you want your identity to be public for this peer review?** For information about this choice, including consent withdrawal, please see our Privacy Policy .

Reviewer #1: No

Reviewer #2: No

**Figure resubmission:**
---

## [Editor Report · Decision Letter 1]

Dear Dr. Martin,

We are pleased to inform you that your manuscript 'Host environment shapes filarial parasite fitness and Wolbachia endosymbionts dynamics' has been provisionally accepted for publication in PLOS Pathogens.

Best regards,

Keke C. Fairfax, PhD

Academic Editor

PLOS Pathogens

Jeffrey Dvorin

Section Editor

PLOS Pathogens

Sumita Bhaduri-McIntosh

Editor-in-Chief

PLOS Pathogens

orcid.org/0000-0003-2946-9497

Michael Malim

Editor-in-Chief

PLOS Pathogens

orcid.org/0000-0002-7699-2064
---

## [Editor Report · Acceptance letter]

Dear Dr. Martin,

We are delighted to inform you that your manuscript, "Host environment shapes filarial parasite fitness and Wolbachia endosymbionts dynamics," has been formally accepted for publication in PLOS Pathogens.

Best regards,

Sumita Bhaduri-McIntosh

Editor-in-Chief

PLOS Pathogens

orcid.org/0000-0003-2946-9497

Michael Malim

Editor-in-Chief

PLOS Pathogens

orcid.org/0000-0002-7699-2064